# Retrospective correction of motion artifact affected structural MRI images using deep learning of simulated motion

**Ben A Duffy**[*]
USC Stevens Neuroimaging
and Informatics Institute

**Wenlu Zhang**[†]
California State University Long Beach

**Haoteng Tang***
USC Stevens Neuroimaging
and Informatics Institute

**Lu Zhao***
USC Stevens Neuroimaging
and Informatics Institute

**Meng Law***
USC Stevens Neuroimaging
and Informatics Institute

**Arthur W Toga***
USC Stevens Neuroimaging
and Informatics Institute

**Hosung Kim***
USC Stevens Neuroimaging
and Informatics Institute
hkim@ini.usc.edu

## Abstract

Head motion during MRI acquisition presents significant problems for subsequent neuroimaging analyses. In this work, we propose to use convolutional neural networks (CNNs) to correct motion-corrupted images as well as investigate a possible improvement by augmenting L1 loss with adversarial loss. For training, in order to gain access to a ground-truth, we first selected a large number of motion-free images from the ABIDE dataset. We then added simulated motion artifacts on these images to produce motion corrupted data and a 3D regression CNN was trained to predict the motion-free volume as the output. We tested the CNN on unseen simulated data as well as real motion affected data. Quantitative evaluation was carried out using metrics such as Structural Similarity (SSIM) index, Correlation Coefficient (CC), and Tissue Contrast T-score (TCT). It was found that Gaussian smoothing as a conventional method did not significantly differ in SSIM, CC and RMSE from the uncorrected data. On the other hand, the two CNN models successfully removed the motion-related artifact as their SSIM and CC significantly increased after their correction and the error was reduced. The CNN displayed significantly larger TCT compared to the uncorrected images whereas the adversarial network, while improved did not show a significantly increased TCT, which may be explained also by its over-enhancement of edges. Our results suggest that the proposed CNN framework enables the network to generalize well to both unseen simulated motion artifacts as well as real motion artifact-affected data. The proposed method could easily be adapted to estimate a motion severity score, which could be used as a score of quality control or as a nuisance covariate in subsequent statistical analyses.

[*]USC Stevens Neuroimaging and Informatics Institute, University of Southern California, CA 90033

[†]Department of Computer Engineering and Computer Science, California State University Long Beach, Long Beach, CA 90840

1st Conference on Medical Imaging with Deep Learning (MIDL 2018), Amsterdam, The Netherlands.

# 1 Introduction

Head motion during MRI scanning has become a critical issue as recent studies rely on 3D acquisition which require longer acquisition times for high-quality imaging, resulting in serious confounding effects for subsequent neuroimaging analyses. Subject motion during scanning results in blurring of tissue contrast boundaries as well as ghost repetitions of the image in the phase-encoding directions. Quasi-periodic motion e.g. due to physiological activity e.g. respiration, results in coherent ghosting artifacts, whereas random motion, manifests as multiple displaced replicas of the image, or stripes [1]. Such neuroimaging confounds become more germane in imaging studies of infants, children [2] and adolescents [3] as they may be less compliant during the imaging session. Consequently, a significant proportion (10-40%) of the initially acquired samples have to be excluded in the analysis stage [4]. Even after exclusion of images with visually-recognized motion artifact through a standard image quality control, the confounding effects of subtle artifacts in the remaining data may be substantial and sufficient to bias results from morphometric studies [5].

Previous attempts to reduce motion artifacts from MRI images are based on iterative estimation of a phase-correction [6, 7], or more recently on compressed-sensing theory [8] or parallel-imaging reconstruction methods [9]. These techniques require the raw frequency domain (k-space) data, which is seldom available for large scale open datasets. For this reason, we adopt a convolutional neural network (CNN) regression model to perform an image domain motion correction based on deep learning.

The CNN training process by nature requires the combination of a set of motion-free images as the ground truth data with the same individual's motion-corrupted images as the input data. It is however impractical to acquire such coupled data. To overcome, we propose a hypothesis that adding realistic motion simulation to clean images and training a CNN model with the combination of clean images and their motion simulation data can correct the simulated motion corruption as well as the real motion artifact. The novelty of our study lies in: (1) selecting a large sample (n=725) of motion-free images as the ground truth, modeling of realistic motion-simulation by applying phase shifts to produce phase discontinuities [7]; (2) Testing the proposed hypothesis by performing a systematic validation and measuring quantitative evaluation indices that have become standard in the field of image reconstruction, and (3) investigating a possible improvement of the regression CNN by adopting an adversarial network approach.

# 2 Methods

## 2.1 Outline

The proposed study framework is described in Fig. 1. It consists of 3 main stages: (1) Training of regression CNN using simulated data; (2) testing using unseen simulated data and the evaluation metrics: SSIM – structural similarity index, correlation coefficient and RMSE; (3) Testing on real motion corrupted volumes using histogram analysis and by measuring TCT – tissue contrast t-score.

## 2.2 Dataset

Seven hundred and twenty five 3D T1-weighted MRI images from the Autism Brain Imaging Data Exchange (ABIDE) dataset 10 that were deemed to have no significant artifacts by our in-house quality control protocol were used for training of the regression CNN. A separate subset of the ABIDE data (n=39) were held out for use as a motion-simulated test set. In the ABIDE dataset which includes a large number of pediatric subjects, we observed 158 cases with a significant amount of motion artifacts (mild: 64; moderate: 84; severe: 20). Twenty-four volumes among the moderately motion-corrupted images were randomly selected to be used for the real data test (note: tissue segmentation error in mild cases were minimal whereas neuroanatomy in the severe cases could be barely be identified). The imaging sequence and acquisition parameters for ABIDE dataset are available from: http://fcon_1000.projects.nitrc.org/indi/abide..

## 2.3 Motion artifact simulation

Motion simulation was performed online during training. Anterior-posterior and left-right axes were considered to be phase-encoding for all simulations. Translational motion can be modeled

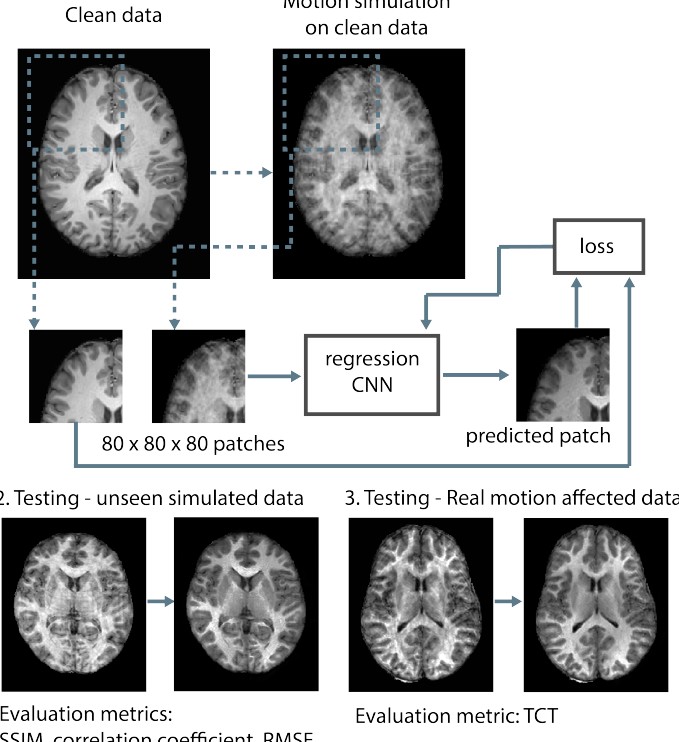

1. Training model - simulated data

Clean data

Motion simulation on clean data

loss

regression CNN

80 x 80 x 80 patches

predicted patch

2. Testing - unseen simulated data

3. Testing - Real motion affected data

Evaluation metrics:
SSIM, correlation coefficient, RMSE

Evaluation metric: TCT

Figure 1: Summary of the proposed study framework. The details of the CNN model are further illustrated in Fig. 2.

as multiplications in k-space by linear phase shifts ($exp(-2\pi i \, k_x \theta_x)$) in the direction of motion, where $k_x$ is the k-space line and $\theta_x$ is a function of the motion. For our purposes, artifacts were simulated by applying random linear phase shifts to n random phase-encoding lines in the Fourier transformed magnitude image, where n was sampled from a uniform distribution between zero and 60. We preserve the center 7 percent of k-space lines as corrupting these would cause low-spatial frequencies to appear dark and thus unrepresentative of most data that would make it to the data archive stage. Ghosting of the bright fat tissue outside of the skull is a common problem that we were able to preserve in our simulations by performing corruption of k-space prior to brain masking and cropping. In a similar manner, the simulated test dataset was created by corrupting 30 lines in the Fourier domain.

## 2.4 Architecture and implementation

Preprocessing consisted of motion simulation (described previously), brain extraction, spatial normalization, cropping and histogram normalization respectively. Brain masks were generated using the Human Connectome project (HCP) pre-processing pipeline [11] which was robust for both motion-free and motion-corrupted images. Masked images were linearly spatially normalized using FSL-FLIRT [12].

CNNs were trained using input patches of size 80 x 80 x 80 voxels using NiftyNet [13] and TensorFlow [14]. A modified HighRes3dNet (HR3DNet) architecture [15], a compact and efficient CNN model suited for large-scale 3D image data was used for regression (Fig. 2) combined with an Adam optimizer [16], an L1 loss function and a batch size of 1. Separately we also used HR3DNetGAN (HR3DNet generative adversarial network) that integrated a discriminator trained to distinguish between clean data and corrected data. Together the generator and discriminator formed an additional adversarial loss term [17] which was added to the L1 loss with a scaling factor of $\lambda$, where $\lambda$ was a hyperparameter set here equal to 0.001. Networks were trained on a single (Nvidia GTX1080Ti,

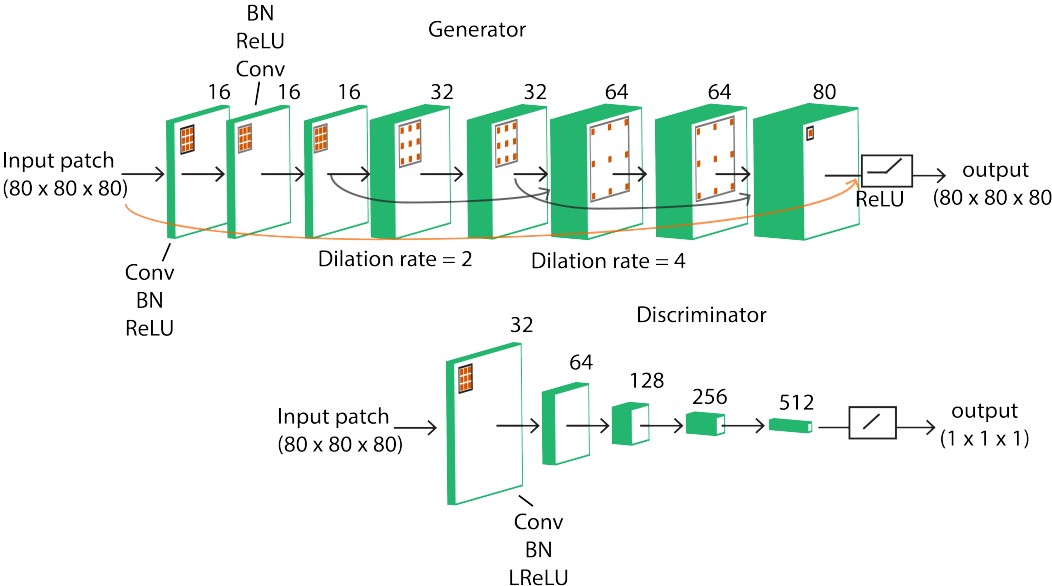

Figure 2: Modified 8-convolutional layer HighRes3dNet [15] architecture with dilated convolutions and skip connections as the generator. The first convolutional layer used convolution, Batch Normalization (BN) then a Rectified Linear Unit (ReLU) activation, subsequent layers used BN, ReLU then Conv. For the discriminator, a 5-convolutional layer network was used with Leaky Rectified Linear Unit (LReLU) as the activation functions in order to avoid sparse gradients.

11GB memory), for 100000 iterations (15 hours) with a learning rate of $10^{-4}$, which was decayed by a factor of 2 every 5000 iterations.

## 2.5 Evaluation

To perform quantitative evaluations on the accuracy of the motion correction using the proposed CNN models, we measured the following metrics that were obtained by comparing the ground truth image and the output image from each CNN model: the structural similarity index (SSIM), Pearson correlation coefficient (CC) and root-mean square error (RMSE). Where SSIM is given by [18]:

$$SSIM = \frac{(2\mu_x\mu_y + c_1)(2\sigma_{xy} + c_2)}{(\mu_x^2 + \mu_y^2 + c_1)(\sigma_x^2 + \sigma_y^2 + c_2)} \tag{1}$$

where $c_1$ and $c_2$ are small constants to stabilize the computation and $\mu_x$ and $\sigma_x^2$ are the mean and variance of the images respectively with $x$ and $y$ indicating the different images to compare. We measured these metrics in the test-set with 39 images with motion simulation. As a 3D Gaussian smoothing has been commonly used to reduce the effect of motion artifact, we assessed the improvement of using CNN models to this conventional method: we compared the SSIM and CC for the results from each HR3DNet and HR3DNetGAN models to those from 3D Gaussian smoothing using a kernel width of 1 mm at full-width half-maximum. To evaluate the performance of the CNN models in correcting the real motion artifact, where the ground-truth was not available, images were classified into GM, WM and CSF using FSL-FAST and tissue contrast t-score (TCT) was used as a surrogate measure of image quality, where TCT was given by [19]:

$$TCT = \frac{(\mu_{wm} - \mu_{gm})}{(\sigma_{wm}^2 + \sigma_{gm}^2)} \tag{2}$$

where $\mu_{wm}$ and $\mu_{gm}$ are the mean white matter and gray matter signal intensities and $\sigma_{wm}^2$ and $\sigma_{gm}^2$ are the variances. The TCT has previously been used to quantify the increase in intensity variation within GM and WM tissue as a result of motion-induced blurring [19].

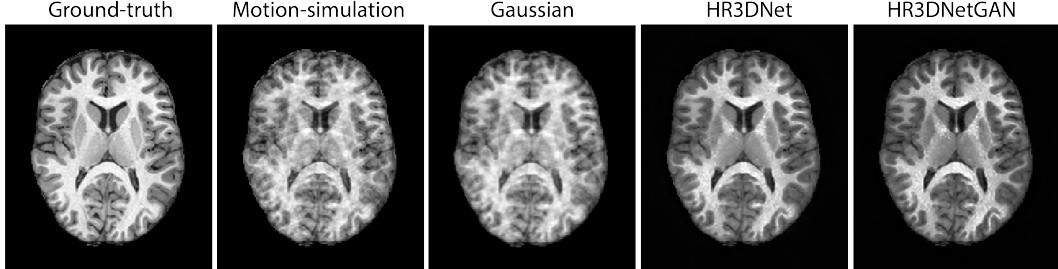

| Ground-truth | Motion-simulation | Gaussian | HR3DNet | HR3DNetGAN |

Figure 3: Examples of testing the CNNs on unseen simulated motion artifacts compared to Gaussian smoothing

Table 1: Quantitative evaluation of model performance on motion artifacts simulated on unseen test cases. Significant increases compared to the uncorrected images is indicated with ** at p<0.0001 (corrected using Bonferroni adjustment)

|  | Uncorrected | Gaussian | HR3DNet | HR3DNetGAN |
|---|---|---|---|---|
| SSIM (Mean $\pm$ SD) | 0.954$\pm$0.0148 | 0.952$\pm$0.0145 | 0.972$\pm$0.009** | 0.971$\pm$0.009** |
| CC (Mean $\pm$ SD) | 0.905$\pm$0.050 | 0.905$\pm$0.050 | 0.960$\pm$0.017** | 0.957$\pm$0.018** |
| RMSE (Mean $\pm$ SD) | 0.0412$\pm$0.010 | 0.0408$\pm$0.009 | 0.0333$\pm$0.011** | 0.0326$\pm$0.011** |

## 3 Results

### 3.1 Simulated test data

We first visually evaluated the image quality for the series of the unseen MRI images used in testing. Our evaluation confirmed that most of the pattern related to simulated motion was well-corrected using both NR3DNet and HR3DNetGAN while maintaining the tissue contrast and sharpness of tissue borders in the corrected images to a level similar to the ground-truth images. Whereas the traditional Gaussian smoothing did not fully remove the pattern related to motion and did not achieve as good tissue contrast while blurring tissue boundaries (example shown in Fig. 3). The quantitative evaluation measures are shown in Table 1. We found that Gaussian smoothing did not significantly differ in SSIM, CC and RMSE from the uncorrected data (p>0.3; paired t-test). On the other hand, the two CNN models successfully removed the motion-related artifact as their SSIM and CC significantly increased after their correction (SSIM: 0.954 vs. 0.972-0.971; p<0.001; CC: 0.905 vs. 0.960-0.957; p<0.001, RMSE: 0.0412 vs. 0.0333-0.0326; p<0.01). The inter-individual variances of SSIM and CC in the CNN corrected images were observed to be 2-3 times lower than the uncorrected data (Table 1).

### 3.2 Real motion-artifact affected data

An example for visual inspection is shown in Fig. 4 and indicates that while some artifacts remain, HR3DNet was able to eliminate a large proportion of ghosting artifacts at the same time as preserving the sharp tissue boundaries and improving the GM-WM tissue contrast. Compared to HR3DNet, use of HR3DNetGAN produces sharper tissue contrast boundaries, however, it tended to over-enhance edges and regions with residual artifacts.

Quantitative results are shown in Table 2. HR3DNet exhibited a significantly larger TCT (p<0.00001) compared to the uncorrected images. The HR3DNetGAN did not show a significant improvement in TCT, which may be explained by its over-enhancement of edges.

Table 2: Quantitative evaluation of model performance on real motion artifact affected data. Significant increases compared to the uncorrected images is indicated with * at p<0.00001 (corrected using Bonferroni adjustment)

|  | Uncorrected | Gaussian | HR3DNet | HR3DNetGAN |
|---|---|---|---|---|
| TCT (Mean $\pm$ SD) | 1.987 $\pm$ 0.117 | 1.905 $\pm$ 0.184 | 2.190 $\pm$ 0.140* | 2.026 $\pm$ 0.135 |

| Original | Gaussian | HR3DNet | HR3DNetGAN |
|----------|----------|---------|------------|

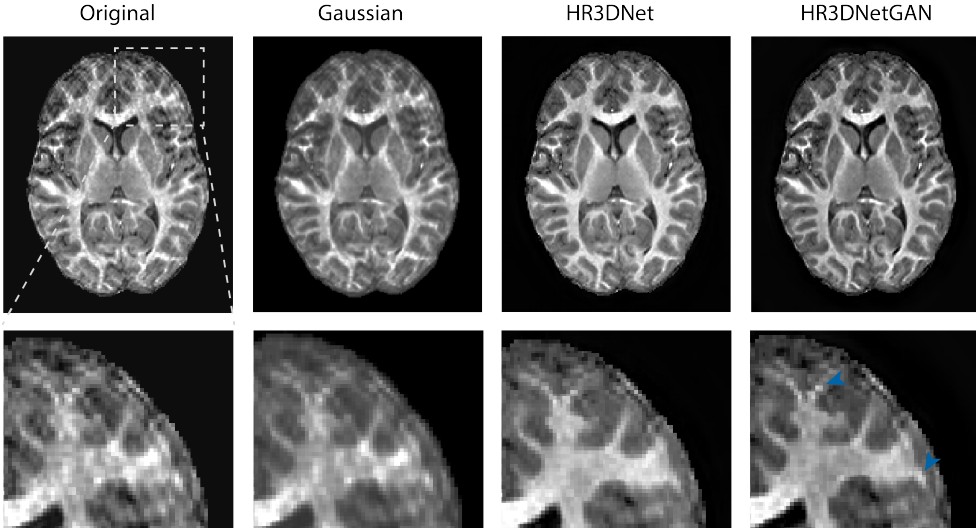

Figure 4: Testing on real motion affected MRI data from the ABIDE dataset. Original motion affected data displays blurring and ghosting artifacts. Gaussian smoothing is able to remove some of the artifact at the expense of blurring and loss of tissue-contrast boundaries. HR3DNet significantly improves the quality of the image, while retaining sharp tissue-contrast boundaries. HR3DNetGAN gives similar results while over-enhancing edges and residual artifacts (blue arrow-heads).

## 4   Discussion and Conclusions

Here we have investigated whether 3D CNNs can be used for retrospective motion correction in the image domain. To this end, we performed a systematic qualitative and quantitative evaluation on simulated and real motion artifact-affected images. Results from this study suggest that training the model on a large database of clean images to define a ground truth enables the network to generalize well to both unseen simulated motion artifacts as well as real motion artifact-affected data which likely present patterns that are not fully characterized using simulations. Our evaluation results however are promising and demonstrate the ability of CNN models trained using simulated data to correct for real motion artifact.

The strength in the proposed approach is that it does not require availability of raw k-space data and could easily be adopted without a notable modification to correct other commonly occurring artifact types such as RF spikes, inhomogeneity, aliasing and Gaussian noise. One early proof-of-principal attempt at using 2D CNNs to correct magnitude domain images suggested a possibility of deep learning-based motion correction [20] but it was difficult to evaluate the performance of this approach as it included neither a quantitative evaluation nor validation on a separate test set with real artifacts. Unlike the CNN model used in [20], recent developments in deep learning, such as residual connections [21], and dilated convolutions [22] that were adopted in our study have enabled us to achieve more promising results for real-world application. In contrast to our hypothesis, although HR3DNetGAN performed correction of real motion artifacts, it also tended to over-enhance edges which confirmed previous findings that GANs tend to introduce artifacts in denoising applications [23]. However, addition of a perceptual loss [23, 24] or using new techniques to stabilize training [25] may help alleviate this issue.

In summary, given the degree of improvement in the quality of motion corrupted MRI data, our method has significant potential to improve performance related to subsequent image processing e.g. brain tissue segmentation. It is practical that such an improvement can be achieved without the need for hand-labelled data, which is opposed to techniques relying on multi-templates and label-fusion. In the image quality control procedure, scoring the severity of the given artifact is a crucial step towards the decision to exclude the artifactual image or not in the subsequent image analysis. Our proposed method could easily be adapted to estimate a motion severity score, for instance, by calculating the evaluation metrics used in our study (i.e., SSIM, CC, TCT) between the original and the corrected

images. These measures could be used as a score of quality control or as a nuisance covariate in subsequent statistical analyses as in [26], which potentially increases the statistical power in the identification of brain changes in relation to neurological conditions.

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
