# OpenReview forum: "Retrospective correction of motion artifact affected structural MRI images using deep learning of simulated motion"
_MIDL.amsterdam/2018/Conference — MIDL 2018 Poster_

### Review · AnonReviewer1 · 2018-05-07
**The paper describes a deep learning approach for motion compensation of structural 3D MR images. While the manuscript describes a valid approach using convolutional neural networks, it would be appreciated if the approach would be better evaluated using real world motion patterns and data.**

**Rating:** 4
**Confidence:** 3

**Review:**

The manuscript describes the use of two convolutional neural network approaches for correction of motion artifacts in structural 3D MR images. This is a worth-while approach, since motion affects interpretability of the images as well as performance of post-processing algorithms. Previous work has dealt with this issue and is not fully cited in this abstract (like Kristof Meding, Alexander Loktyushin, Michael Hirsch: Automatic detection of motion artifacts in MR images using CNNS, ICASSP 2017 - 2017 IEEE International Conference on Acoustics, Speech and Signal Processing (ICASSP), which use a similar approach). The paper is clearly written and easy to understand. The quality is mixed. While the authors seem to have a good understanding of machine learning approaches and similary measures they seem to lack understanding of MRI. Simulated motion artifacts are only partially realistic.

The example with real-world motion (fig.4) seems to be a good pick, which worked well. Nonetheless, both CNN models remove a lot of information relevant for diagnostic purposes from the white matter signal.

It also remains unclear how the performance of the described approach would be in case of strong motion.

At which step was the segmentation of grey and white matter performed for the real motion affected MRI data? Before or after motion compensation? How meaningful would this be?

Pros:
- use of 3D patches instead of 2D
- nice use of adversarial CNN

Cons:
- use of magnitude images doesn't allow reconstruction of k-space, thus motion simulation is somewhat unrealistic
- used motion patterns (translation within axial plane) is not realistic. Often, nodding motion will occur, which is not captured in this approach. The examples, however, show mild rotational motion around z-axis ("no" motion), which can be easily approximated by inplane translation.
- selection of similarity measures does not show better (?) performance of the HR3DNetGAN approach.


**Special Issue:**

Yes

---

### Review · AnonReviewer2 · 2018-05-10
**Application of 3d CNN and GAN for motion estimation and artifact correction, clearly written and well organized, Evaluation done on simulated data, Discussion should include clinically significant information loss; Borderline Accept**

**Rating:** 3
**Confidence:** 2

**Review:**

The proposed methodology illustrates the application of two 3D convolutional neural network for motion artifacts estimation and correction in structural 3D MR images. The methodology is quantitatively and qualitatively evaluated in an appropriate manner.

What type of clinically significant information is lost during reconstruction? Some discussion would be useful.

Pros:
- Use of 3D CNN and GAN for motion correction
- Clearly written

Cons:
- It is not clear how the proposed approach differs from the one published in "Automatic detection of motion artifacts in MR images using CNNS, ICASSP 2017".
- The methodology is evaluated on simulated data. How well does the simulated data reflect real life motion artifacts.


**Special Issue:**

No

---

### Review · AnonReviewer3 · 2018-05-10
**This paper investigates the use of GANs and CNN regression for motion correction of brain images.**

**Rating:** 4
**Confidence:** 2

**Review:**

The author presents a novel motion correction method for brain images using 3D CNN based regression. The method is novel, the results are promising and comparison to GANs is interesting. Overall the paper is well illustrated and written. Some additional validations would make the paper even better

1.  How does the model compare to traditional registration based methods like Nifty-Reg/ANTs for motion correcttion?
2.  What is the error metric for tissue similarity for 3D CNN regression model compared to a template brain image selected from the training image?

**Special Issue:**

Yes

---

### Comment · ~Soumya_Ghose1 · 2018-05-05
**This paper investigates the use of GANs and CNN regression for motion correction of brain images. Recommendation: Accept.**

Recommendation : Accept.

The author presents a novel motion correction method for brain images using 3D CNN based regression. The method is novel, the results are promising and comparison to GANs is interesting. Overall the paper is well illustrated and written. Some additional validations would make the paper even better

1.  How does the model compare to traditional registration based methods like Nifty-Reg/ANTs for motion correcttion?
2.  What is the error metric for tissue similarity for 3D CNN regression model compared to a template brain image selected from the training image?

---

> ### Comment · ~Ben_A_Duffy1 · 2018-05-10
> **Thank you for the comments**
>
> Dear Soumya Ghose, thank you for your interest in our work, for the generous comments and stimulating discussion. We have discussed your ideas below.
>
> 1.	How does the model compare to traditional registration based methods like Nifty-Reg/ANTs for motion correction?
> The registration-based methods are appropriate for datasets where there exist multiple frames e.g. fMRI or DTI (presumed to have no 3D motion in a frame volume but have between-frames misalignment) or between-slices misalignment due to the motion involved in 2D image acquisition and lack the capacity to correct the 3D motion artifact involved in a 3D image acquisition. Thus, these methods would not be suited to correction of the motion that occurs during acquisition of a single structural MRI volume, as the artifacts in this case generally result in blurring and ghosting which could not be alleviated using registration-based methods. We would have liked to compare against reconstruction-based techniques, however these are only really feasible when the raw k-space data are available.
>
> 2.     What is the error metric for tissue similarity for 3D CNN regression model compared to a template brain image selected from the training image?
> We measured the Tissue Contrast T-score (TCT) for the evaluation of real motion artifact-affected images because the ground-truth was not available for these data. Whereas we evaluated the simulated motion using RMSE, CC and SSIM, where the ground-truth (i.e., image with no motion simulation) was present and the clear improvements in quality were obvious by the assessment using these metrics. We thus had not considered using TCT for the simulated data.  Nevertheless, what you suggest is an excellent idea and we really appreciate the feedback. For the full paper we will further validate the tissue similarity by comparing the tissue-contrast values from the motion-simulated data, corrected data and the ground-truth (i.e., the template image with no motion).

---

### Comment · ~Bram_van_Ginneken1 · 2018-05-18
**Selection for longlist for special issue Medical Image Analysis**

Dear authors,

Congratulations on your acceptance to MIDL! We have selected your paper on the longlist for the Medical Image Analysis Special Issue. Please read this page:
https://midl.amsterdam/special-issue-in-medical-image-analysis/
Please answer the three questions that are listed on that page about your interest in submitting to the special issue, potential overlap with other publications, and related publications.

You can post your answer here directly below on openreview.net, or mail me directly at bram.vanginneken@radboudumc.nl.

Best regards, Bram

---

### Decision · Program_Chairs · 2018-05-15
**Paper94 Acceptance Decision**

Poster